# Compound Haplotype Variants in *CFH* and *CD46* Genes Determine Clinical Outcome of Atypical Hemolytic Uremic Syndrome (aHUS)—A Series of Cases from a Single Family

**DOI:** 10.3390/jpm11040304

**Published:** 2021-04-15

**Authors:** Agnieszka Furmańczyk-Zawiska, Anna Kubiak-Dydo, Ewelina Użarowska-Gąska, Marta Kotlarek-Łysakowska, Katarzyna Salata, Monika Kolanowska, Michał Świerniak, Paweł Gaj, Beata Leszczyńska, Maria Daniel, Krystian Jażdżewski, Magdalena Durlik, Anna Wójcicka

**Affiliations:** 1Department of Transplantation Medicine, Nephrology and Internal Diseases, Medical University of Warsaw, 02-006 Warsaw, Poland; afurmanczyk@gmail.com (A.F.-Z.); magdalena.durlik@wum.edu.pl (M.D.); 2Warsaw Genomics INC, 01-682 Warsaw, Poland; kubianka.anka@gmail.com (A.K.-D.); ewelina.uzarowska@warsawgenomics.pl (E.U.-G.); marta.kotlarek-lysakowska@warsawgenomics.pl (M.K.-Ł.); kasia.falana@gmail.com (K.S.); monika.kolanowska@warsawgenomics.pl (M.K.); michal.swierniak@warsawgenomics.pl (M.Ś.); pawel.gaj@warsawgenomics.pl (P.G.); krystian.jazdzewski@warsawgenomics.pl (K.J.); 3Postgraduate School of Molecular Medicine, Medical University of Warsaw, 02-091 Warsaw, Poland; 4Department of Pediatrics Nephrology, Medical University of Warsaw, 02-091 Warsaw, Poland; bleszczynska@wum.edu.pl (B.L.); maria.daniel@wum.edu.pl (M.D.); 5Laboratory of Human Cancer Genetics, University of Warsaw, 02-089 Warsaw, Poland

**Keywords:** aHUS, hemolytic-uremic syndrome, CFH, CD46, kidney transplantation

## Abstract

Atypical hemolytic uremic syndrome (aHUS) is a rare disease triggered by dysregulation of the alternative complement pathway, consisting of a characteristic triad of nonimmune hemolytic anemia, thrombocytopenia, and renal failure. The risk of aHUS onset, recurrence, and allograft loss depends on the genetic background of a patient. We show a series of cases from a single family whose five members were affected by aHUS and presented distinct clinical outcomes. Next-generation sequencing revealed combined mutations in both complement factor H and membrane cofactor protein CD46. Out of eight siblings, aHUS affected three adult brothers, and, subsequently, affected two children of an unaffected sister. The first patient died due to aHUS, and two other brothers underwent successful kidney transplantation with no aHUS recurrence. The younger, 10-month-old child presented with a severe course of the disease with cardiac involvement and persistent hemolytic anemia limited by eculizumab, while the 2-year-old recovered completely on eculizumab. The study shows a highly variable disease penetrance.

## 1. Introduction

Hemolytic uremic syndrome (HUS) is a rare disease consisting of the characteristic triad of clinical symptoms: nonimmune microangiopathic hemolytic anemia (MAHA) thrombocytopenia, and acute kidney injury (AKI) [1,2]. While 90% of all cases are classified as a typical HUS, caused by Shiga toxin producing Escherichia coli infection, 10% are related to the genetic or acquired alternative complement pathway dysregulation, and are known as atypical HUS (aHUS). The onset of aHUS ranges from childhood (41.6%) to adulthood (58.4%) [3]. Numerous variants in complement factor H (*CFH*), membrane cofactor protein (*MCP/CD46*), complement factor I (*CFI*), complement factor B (*CFB*), CFH related (*CFHR1, CFHR2, CFHR3, CFHR4*), and diacylglycerol kinase epsilon (*DGKE*) genes, predisposing to aHUS, have been identified. Although the genetic variants associated with aHUS are relatively common and occur in up to 30–40% of the general population, clinical presentation of the phenotype and severity of the disease depend on the culpable genetic background. About 50–60% of genetic aHUS cases progress to end-stage renal disease (ESRD). The risk of aHUS recurrence and allograft loss depends on the complement genetic alterations. Genetic diagnostics prior to qualification for kidney transplantation is recommended [4,5].

A wide spectrum of genetic variants in the complement system is reflected by a variety of clinical presentations of the disease. Depending on the mutation type, gene penetration, and the presence of combined mutations (at least 2 pathogenic variants), the clinical course may differ, including a single sporadic, acute episode or a recurrent or familial disease. CFH is the most important regulator of complement alternative pathway and controls complement activation in the fluid phase as well as on the cell surfaces. Genotype/phenotype correlations in genetic variability within the *CFH* may cause several diseases depending on location of variants in the *CFH* functional region. *CFH* consists of 20 short consensus repeats (SCR) and mutations located in the C-terminal region (SCR 19–20) are associated with aHUS risk haplotype, whilst aberrances in the N-terminal region (SCR 1–4) are associated with C3-glomerulopathy risk haplotype [6,7,8,9,10]. In case of genetic alterations located in *MCP/CD46* gene or deletion of *CFHR* cluster, the recurrence risk is lower than 20%; thus, renal transplantation is generally an accepted approach. In case of genetic variants in the *CFH, CFI,* or *CFB* genes, the risk of disease recurrence in the transplanted organ exceeds 80% [11]. The incidence of combined mutations in the Italian cohort was 3.4%; 8–10% of patients with *CFH, C3,* or *CFB* had combined mutations, whilst combined mutations in *MCP*- or *CFI* were observed in 25% of the patients [12]. In the French cohort, combined mutations were found in 4.2% of aHUS patients (children and adults) [13]. Analysis of asymptomatic family members of aHUS patients revealed that the genetic disturbances in the complement-related genes existed in approx. 50% of analyzed persons, indicating complex background of the disease and a multitude of factors that trigger its symptoms.

In this paper, we describe a series of cases from a single family with a clinical follow-up of 14 years, in which five members were affected by aHUS and presented distinct clinical outcomes. Next-generation sequencing revealed complex genetic background and familial aggregation of the disease.

## 2. Materials and Methods

Among eleven family members, five were clinically diagnosed with aHUS (Figure 1). In this study, aHUS affected three brothers with adult onset of the disease and two daughters of an unaffected sister with infant onset of the disease. The affected children were treated with eculizumab—the recombinant humanized monoclonal antibody that inhibits the terminal complement alternative pathway. The remaining family members never presented with the disease symptoms. There was also no consanguinity. This retrospective study was performed in concordance with the local Ethical Committee at the Medical University of Warsaw. Informed consent for genetic testing was obtained from each of the enrolled subjects or legal guardians in case of minors. 

CFH-Ab analysis: The presence of anti-CFH antibodies was assessed with the ELISA-VIDITEST assay (VIDIA), according to the manufacturer’s protocol. The results were compared with the reference value for the Polish population obtained at Warsaw Genomics, which ranges from 9.44 to 15.22 AU/mL.

Next-generation sequencing: DNA was extracted from 9 mL of peripheral blood by the salting out method. The quality of nucleic acids was validated by spectrophotometer on NanoDrop 2000 (Thermo Scientific) and quantity on the Quantus (Promega) fluorometer with the QuantiFluor dsDNA System (Promega). The minimum quality parameters of isolated DNA were 1.8 for 260/280 and 260/230 ratios. The integrity of genomic DNA was confirmed in electrophoresis. Libraries were prepared with the use of the SeqCap EZ Choice Enrichment Kit (Roche) and the analyzed genes included *ADAMTS13; C3; CD46; CFB; CFH; CFHR1; CFHR2; CFHR3; CFHR4; CFHR5; CFI; DGKE; THBD*. The library included the coding regions of the genes together with the intronic variants within the CD46 and *CFH* risk haplotypes (rs3753394, rs3753396, rs1065489, rs2796267, rs2796268, rs1962149, rs859705, rs7144). A measure of 1 µg of gDNA was sheared by M220 focused-ultrasonicator (Covaris). The average fragment sizes between 150–500 bp were verified on the 2200 Tapestation nucleic acid system (Agilent) and the library was prepared according to the manufacturer’s protocol. The quality and size of the libraries was evaluated on the 2200 Tapestation and the quantity on Quantus. Once passing the quality control, the libraries were pooled and proceed for targeted hybridization. The efficiency of hybridization was verified using real-time PCR (LightCycler 480, Roche). The final library size and quality was validated on 2200 Tapestation and the quantity was measured on Quantus. The sequencing was run as paired-end, 2 × 75 bp on NextSeq 500 sequencing system (Illumina).

Next-Generation Sequencing Data Analysis: NGS data storage and analysis were performed on dedicated computing resources maintained by the Warsaw Genomics at University of Warsaw. Sequencing data were archived as bcl and fastq files formats. The workflow for variant calling was based on open-sourced tools. Trimming adaptors and filter quality of the fastq files were performed with Trimmomatic 0.35 and FastQC algorithms [14,15]. Reads were mapped using Burrows–Wheeler Alignment against human reference genome GRCh37/hg19, duplicates were removed by Picard, realignment was performed with ABRA, and variant calling were done with BCFtools [16,17,18]. Variant annotation was run in few steps using various data collecting tools (SnpSift, SnpEff, VariantEffect) [19,20].

Copy number variation analysis: the presence of large genomic aberrations (deletions, rearrangements) within the *CFH, CFHR1, CFHR2, CFHR3,* and *CFHR5* genes was analyzed using the multiplex-ligation probe amplification system, MLPA P236-A (MRC Holland), according to the manufacturer’s protocol. The products were resolved on an ABI3100 Genetic Analyzer (Life Technologies), and analyzed with GeneMapper software (Life Technologies). In each MLPA run, we included two control samples.

## 3. Case Descriptions

Three brothers developed aHUS in their 20s. Two of them underwent successful kidney transplantation (Ktx) in 2010 (P3) and 2011 (P2), but due to the lack of directed genetic testing, the screening for complement mutations had not been performed prior to Ktx. No other disease run in the family, there was also no precipitating factor identified.

Brother 1 (P1) fell ill in 1999, aged 22. He developed fever, abdominal pain, general malaise, vomiting, and was given an empirical antibiotic therapy for one week with no response. Oliguria appeared. On admission to the local hospital, he was in poor general condition, his blood pressure was 170/100 mmHg. Laboratory findings revealed severe hemolytic anemia: hemoglobin concentration (HGB) was 69 g/L, and hematocrit (HCT) was 18%. The platelets count (PLT) was 72 G/L; white blood count (WBC) was 13.2 G/L; lactate dehydrogenase activity (LDH) was 4936 U/L (range 80–248); anisocytosis with macrocytosis, microspherocytes, and schistocytes were present in the peripheral blood smear; and the direct Coombs’ test was negative. Serum creatinine concentration (SCr) was 7.8 mg/dL (range 0.6–1.3), urea 112 mg/dL (range 20–40), urine analysis revealed active urine sediment with leucocyturia, erythrocyturia, and proteinuria was 650 mg/dL in a portion. Blood transfusion and fresh frozen plasma (FFP) were given. Genetic analysis considering complement abnormalities was not performed at that time. Hemolysis was terminated within few days, with HGB and PLT raised and stabilized. However, there was no renal improvement, and anuria remained, so the patient required hemodialysis. A few days later, the patient deteriorated, lost consciousness, and was referred to intensive care unit. During the hemodialysis session, before therapeutic plasma exchange (TPE) had been introduced, the patient suffered from heart arrest with no response to resuscitation and died. 

In 2006, the 27-year-old second brother (P2) suddenly developed symptoms similar to those presented in the first brother. Local hospital laboratory findings revealed MAHA and AKI. Based on family history, aHUS had been suspected and the patient was referred to the Department of Transplantation Medicine, Nephrology, and Internal Diseases, Medical University of Warsaw. On admission, the patient was conscious, anuric, dialysis-dependent, and hypertensive—his blood pressure was 190/130 mmHg; fever was still present. ADAMTS-13 activity was normal, serum concentration of complement component 3 (C3) and C4, antinuclear antibodies (ANA), antibodies directed against glomerular basement membrane (anti-GBM), anticardiolipin antibodies (ACL) IgG and IgM, and lupus anticoagulant (LA) were within normal limits, LDH 1524 U/L, and serum haptoglobin concentration (HPT) was undetectable. High blood pressure was reduced using standard antihypertensive drugs. Spring-loaded needle kidney biopsy was performed. Two samples of kidney cortex were collected (25 glomeruli) and histopathological examination revealed severe thrombotic microangiopathy (TMA) and acute tubular injury; immunofluorescence examination (IFL) showed IgM, complement, and fibrinogen positive staining in the arterioles. Based on the kidney biopsy, the TPE session was started immediately. Daily TPE sessions were performed in the morning using FFP as a substitute. Moreover, one unit of FFP was given extra in the evening due to ongoing hemolysis. After 12 TPE sessions, hemolysis was limited, but the patient was still anuric and required hemodialysis. In order to assess further prognosis, the second kidney biopsy was performed 3 weeks later. Two samples were collected—in one sample only necrotic scarring was observed, and in the second sample, the kidney cortex contained 9 glomeruli with slightly active TMA and progression of chronic lesions–chronic ischemic nephropathy; IFL revealed fibrinogen positive staining in arterioles. Next, five TPE sessions were performed with slow improvement in urine output (2000 mL/24 h) with no reduction in SCr. Hematological remission was finally achieved; however, the patient remained dialysis-dependent. Severe course of the disease and the risk of aHUS recurrence kept the patient P2 reluctant to pretransplant assessment, so hemodialysis was continued. 

In 2009, the third 27-year-old brother (P3) suffered from isolated hematuria that subsided spontaneously within one day. General malaise and oliguria appeared few days later. Local hospital laboratory findings revealed MAHA and AKI and the patient was referred to the Department of Transplantation Medicine Nephrology, and Internal Diseases, Medical University of Warsaw. On admission, he was in a good general condition. Physical examination found no other abnormalities. ADAMTS-13, C3, and C4 were normal. The panel of antibodies: ACL, LA, ANA, and anti-GBM, were negative. Patient required hemodialysis. During implementation of dialysis, catheter massive bleeding occurred so transfusion of PLT, so FFP and 2 blood units were provided. Due to severe thrombocytopenia and propensity for bleeding, based on clear evidence of family history, kidney biopsy was not taken into consideration. Immediately, daily TPE sessions were started using FFP in the dose of 40 mL/kg as a substitute. Within 2 weeks of treatment, hematological remission was achieved with no renal response. The patient remained dialysis-dependent and was referred to a local dialysis center. Three months later, the patient was admitted again for pretransplant assessment. Despite the risk of aHUS recurrence, he was highly motivated to be accepted as a potential renal transplant recipient. Standard evaluation revealed no absolute contraindications for Ktx. In those days, the crucial problem was in the lack of genetic testing identifying complement abnormalities. Regarding economic reasons, it was not possible to perform it abroad. The patient accepted the risk of aHUS recurrence following kidney transplantation (Ktx). He underwent Ktx from a 60-year-old cadaveric female donor (5 November 2010), and the historical panel reactive antibody (PRA) and PRA prior to Ktx was 0%. In analyzing the immune selection between donor and recipient, three human leucocyte antigen mis-matches (HLA MM) were found. The standard immunosupressive regimen was implemented: glucocorticosteroids, tacrolimus, and mycophenolate mofetile (MMF). The immediate graft function was observed. Within the first 3 months, doses of glucocorticosteroids and tacrolimus (through level 5–7 ng/mL) were tempered and SCr stabilized at 1.3 mg/dL. In the third month after Ktx, in protocol graft biopsy, we found no signs of TMA, no acute rejection, and IFL and C4d staining were negative. His uneventful posttransplant outcome made the second brother (P2) encouraged to try Ktx. The P2 underwent Ktx from a 24-year-old cadaveric female donor (9 September 2011), the PRA was 0%, and there were two HLA MM. The immunosupressive regimen consisted of glucocorticosteroids, tacrolimus, MMF, and basiliximab as per center protocol. Immediate graft function was observed with SCr 1 mg/dL within the first few days. Three months later, protocol graft biopsy revealed normal graft structure, no sign of TMA, no acute rejection, and IFL negative. Over the years (until December 2020), both recipients have remained stable with no signs of aHUS recurrence. Genetic analysis identifying complement abnormalities was conducted in 2016 when such opportunity occurred. Moreover, the P2 decided to procreate, so to avoid potential teratogenic effect, MMF had been switched into azathioprine three months prior to planned conception. In 2017, a healthy infant was born with no sign of the disease so far. In P3, renal cell carcinoma (RCC) of the native kidney was diagnosed in 2019, so he underwent nephrectomy and nephron-sparing surgery for suspicion of allograft RCC. Histological examination revealed papillary RCC type 1, G1 (Fuhrman) pT1a in native kidney, in allograft—a cyst with no sign of tumor.

In 2016, aHUS affected a 2-year-old daughter (P4) of an unaffected sister (S3). The 10-month-old girl was admitted to the Department of Pediatrics Nephrology, Medical University of Warsaw with clinical features of AKI. The symptoms included diarrhea with 4–5 loose feces per day without blood or mucus and vomiting. On admission, she was in moderate general condition with pale skin and peripheral oedema. Weight on admission was 7530 g (<3 percentile) and that was 730 g more than her estimated dry weight. Physical examination revealed abdominal bloating, hepatomegaly, and severe hypertension (BP 150/90 mmHg). Laboratory tests confirmed MAHA and AKI. Urinalysis revealed proteinuria 4894 mg/dL and erythrocyturia. The direct Coombs’ test was negative, decreased C3 was 68.5 mg/dL with normal C4, and the total hemolytic complement measurement (CH50) was 18 µEq/mL (range 70–180). The peripheral blood smear revealed anisocytosis with macrocytosis, microspherocytes, schistocytes, rouleaux, and platelets anisocytosis. Normal ADAMTS-13 activity excluded thrombotic thrombocytopenic purpura (TTP), the fluorescein-labeled proaerolysin (FLAER) test excluded paroxysmal nocturnal hemoglobinuria (PNH), and no warm or cold autoantibodies were found. Concentration of homocysteine and cobalamin were normal. Stool culture was negative. Because of anuria, continuous veno-venous hemodiafiltration (CVVHDF) was initiated (eighteen 24 h long dialysis). Plasma infusions were started (10 mL/kg/day) prior to TPE. Eight TPE sessions were performed and hemolysis was terminated. After the second TPE, transient restitution of diuresis (3 mL/kg/h) was noted; however, three days later, anuria recurred. Five weeks later, the method of renal replacement therapy was changed for peritoneal dialysis. Hemolysis was terminated with no renal improvement. Genetic analysis of complement regulators revealed a combined mutation (Table 1). At that time, eculizumab therapy was not available in Poland.

In the third week of hospitalization, pulmonary oedema was observed. Despite solute removal during CVVHDF, no improvement in left ventricular systolic function or decrease in severity of mitral regurgitation was observed. Ejection fraction (EF) remained around 50%, which may indicate direct cardiac involvement. 

In the following 12 months, the patient had been undergoing peritoneal dialysis and had been nourished by percutaneous endoscopic gastrostomy (body weight–7.6 kg). The laboratory tests constantly showed PLT 234 G/L, HGB 98 g/L (despite of erythropoietin injections). The patient presented with anuria, LDH 800 IU/mL and persistently low C3 levels. The left ventricular function deteriorated and after a year from the onset of aHUS, the ejection fraction (EF) was 28% (EF Simpson-Method).

In 2017, she was vaccinated against meningococcal infection (Men A, C, W-135, Y, and B) and due to severe cardiomyopathy, eculizumab therapy was started (from social donations). Eculizumab blocks the terminal complement alternative pathway (anti-C5 agent) and it is highly effective in complement-mediated TMA. During the first year of eculizumab treatment, a gradual improvement in cardiac EF (% EF Simpson-Method-from 28 to 57%) was observed. In the second year of therapy, a decrease of EF was (50%) observed again and pulmonary hypertension was diagnosed based on cardiac catheterization. Sildenafil was introduced with satisfactory effect. She has not presented cardiovascular insufficiency. Despite a balanced diet and feeding by percutaneous endoscopic gastrostomy, her weight gain and height are unsatisfactory (<3c). The patient is currently waiting for Ktx.

In 2018, aHUS affected the second daughter (P5) of the same unaffected sister (S3). The 4-year-old girl was admitted to the Department of Pediatrics Nephrology with gross hematuria. The symptoms included vomiting, fever, and stomach pain. On admission to the hospital, she was in moderate general condition and physical examination revealed pale skin, petechiae, and middle dehydration. Laboratory tests confirmed MAHA and AKI. Urinalysis revealed proteinuria 1487 mg/dL and erythrocyturia. The direct Coombs’ test was negative, slightly decreased C3 was 68.6 mg/dL with normal C4, and CH50 was 5 µEq/mL (range 70–180). ADAMTS-13 activity and concentration of homocysteine were normal. Shiga toxins was not detected by PCR and stool culture was negative. Due to family history with aHUS, the girl was qualified for the eculizumab treatment program already introduced in 2018. She was vaccinated against meningococcal infection (Men A, C, W-135, Y, and B) and on the third day of hospitalization, the first dose of eculizumab was administered. After a few days, PLT normalized, and decreased renal function parameters and LDH were observed. 

In the following 18 months, the patient has been undergoing eculizumab therapy, and this treatment is still being continued. P5 is now a healthy girl with normal renal function and normal blood pressure, and no recurrence of HUS has been observed.

## 4. Genetic Analysis Reveals Complex CD46/CFH Haplotypes

Both adult patients were homozygous for the CD46ggaac haplotype, while patients with infant onset of the disease were heterozygous for most of the variants. P4 was heterozygous for rs7144 and rs1962149, while P4 was a reference homozygote for rs7144 (Table 1).

In addition, the patients had variants in the CD46 gene, which were not a part of the CD46ggaac haplotype. Only P4 was heterozygous for all these variants, whereas other patients were homozygous (rs4844390, rs11118580) or reference homozygous (rs2724374). Variant rs2724374 was also heterozygous and was previously described in the aHUS patient with adult onset of the disease [21]. The patients had the same zygosity of variants in CFH(H3) haplotype—they were heterozygous for variants rs1065489, rs3753394, rs3753396, and rs800292 and homozygous for rs1061170. The patients were homozygous for the c.921A>C (rs1061147) and c.1204C>T (rs1061170) variants in CFH, which are associated with end-stage renal disease [22]. 

The patients were also heterozygous for the C3 c.941C>T (rs1047286) and c.304C>G (rs2230199) variants, which are associated with increased activity of the alternative complement pathway [23]. Interestingly, the C3 gene contains 10 variants for which the P2 and P3 had the same zygosity, which differed in P4 and P5.

P4 was homozygous for rs11569511 and heterozygous for rs7951, rs389404, rs2230204, and rs2547438, while P5 was homozygous for rs189367 and heterozygous for rs408290, rs433594, and rs748011345. These variants were not detected in both adult patients. 

Other C3 variants have been identified as homozygous in the P2 and P3 patients and heterozygous in the P4 and P5 patients. However, the frequency of the reference allele of these variants is lower than of the alternative allele (MAF). 

Moreover, the infant patient (P4) was heterozygous for both CD46ggaac and CFH(H3) haplotypes, although the clinical outcome of the disease was the most severe among all the affected patients. This patient was also homozygous for the rs2301612 ADAMTS13 variant, also reported in aHUS patients and associated with slight changes in the secretion and the activity of the enzyme (95% of secretion, 75% of activity) in in vitro study [24].

P5 was also homozygous for the rs2301612 ADAMTS13 variant; in addition, there are homozygous variants that have not been detected in other patients.

Genetic analysis revealed no pathogenic variants in the DGKE gene and lack of genomic aberrations within the CFHR1/3 cluster. In concordance with these findings, the anti-H antibodies were within the normal range. However, the affected child is a carrier of a heterozygous variant of unknown significance (17_54926020_T>A, rs1457404667) in the DGKE gene. The analysis also revealed that the healthy father of both adult patients is homozygous for the CFH risk haplotype, although never presented with the disease symptoms. Sister 3, the mother of P4 and P5, has an identical genotype to her affected daughter P4 and never presented with the disease symptoms. Detailed results can be found in the Appendix A.

## 5. Discussion

Over the past decade, there has been a great advance in the understanding of the complement system. The progress in the field of genetic analysis and the introduction of NGS methods made it possible to examine whole complement pathways involved in a specific disease process, without the need of limiting the analysis to single variants of a single gene. Combined complement gene mutations in aHUS deteriorate clinical outcome [13]. The mutations are mostly inherited from unaffected parents, and de novo mutation occurs rarely [25], which was also shown in our study. Incomplete genetic penetrance of the disease (50–60% in aHUS) in carriers with complement gene mutation, deletion, or common at-risk polymorphism is a frequently observed feature. The penetrance increases with the number of aHUS mutations. Although genetic testing has become available, genetic counselling still remains difficult due to plenty of reasons (incomplete penetrance of aHUS variants, novel mutation, protective haplotype, and impact of environmental complement amplifying conditions in individual cases). Analysis of familial pedigree demonstrated that aHUS patients were found to have inherited the complement defect from unaffected parents. That indicates genetic variants affecting the complement are a modifier and a potential precipitating factor is needed to develop clinically overt aHUS [8]. The age-relatedness of aHUS was also debated. The asymptomatic father of adult patients is homozygous for the CFH risk haplotype. He lives with his sons in the same household and probably was exposed to the same trigger (if any) of aHUS as his sons but remains clinically silent. The asymptomatic S3 has an identical genotype to P4 and never presented with the disease. Apart from the triggering factors, viable penetrance that may influence clinical outcome more than age is also emphasized [26]. In our study, we present cases of familial aHUS with a combined CFH/CD46 mutation. In adult patients (a single episode of the disease in adulthood, hematological remission due to TPE without renal improvement, uneventful post-transplant period with no recurrence), prior to genetic analysis, CD46 mutation was suspected. In a French cohort of children [13], the prognosis of patients with a CD46 mutation was relatively better when compared to other mutations. CD46-mutated patients rarely developed ESRD at first presentation [22]. The risk of recurrence after Ktx is also low in such cases [25,27]. MCP mutation is associated with longer time to ESRD than CFH mutation [20]. However, in our study, we observed AKI and anuria with no response to TPE, which showed previously in case of combined mutation [13,27]. Contrary to our findings [28], recently published data derived from The Global aHUS Registry shows that aHUS patients with initial presentation in childhood are more likely to have MCP or CFI mutation. Similar to others [7], we also could not identify a potential precipitating factor in adult patients; in the case of an affected child, diarrhea could be considered as such a condition [20].

According to previously published data [5,13,28], double-mutated patients are at significantly higher risk of ESRD at onset or first year (30–40%). In combined MCP mutations (plus CFH or CFI or C3), the risk of recurrence in allograft is 50–60%, whilst in isolated MCP mutation, the risk of recurrence in allograft is estimated to be less than 20%. Transplant outcome in aHUS recipients with combined mutations was 44–50% of renal graft loss within 3 years [13]. In our adult patients, over 10 years of post-transplant follow-up, we observed stable renal graft function with 1 and 1.3 mg/dL SCr in P2 and P3, respectively. In P2, daily proteinuria is maintained at less than 0.5 g on 3 antihypertensive drugs, and P3 has no proteinuria. Over ten years of post-transplant follow-up, we did not observe signs of recurrence in P2 and P3; P4 is under the process of listing for transplantation with eculizumab. The anti-C5 therapy with eculizumab is a well-known and effective management of aHUS in transplant recipients. However, the optimal length of treatment remains unknown. Nowadays, TPE in aHUS is considered as a bridging therapy and eculizumab is the first line treatment. This study shows an example of a compound aHUS background and different disease outcomes within a single family. The limitations of the study are its retrospective nature and lack of genetic testing in P1. To our knowledge, there are scarce data considering a series of cases from a single aHUS family with a long posttransplant follow-up. The distinct clinical outcome of the insidious disease and the effective treatment are the strengths of the study. The use of Next Generation Sequencing, a comprehensive method of multigenic analysis, made the analysis more precise and complex. 

## Figures and Tables

**Figure 1 jpm-11-00304-f001:**
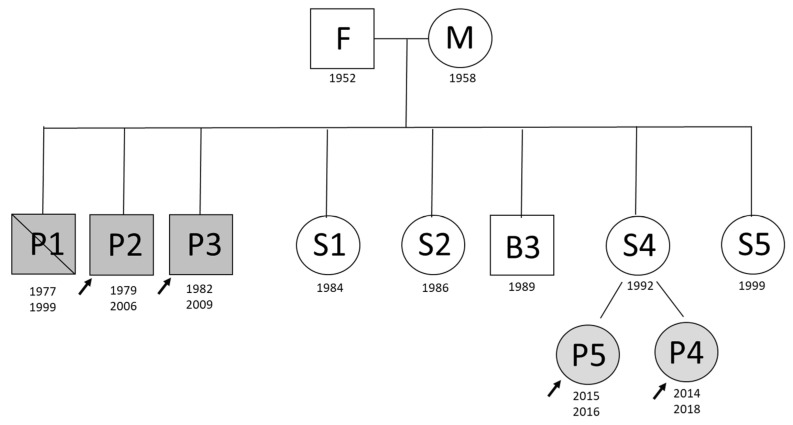
Pedigree of the aHUS-affected family. The deceased individual (P1) is depicted with diagonal line through it. Under each individual, the year of birth and, if appropriate, the individual’s year of disease onset appear. Black colored shapes indicate aHUS patients. The patients described in the study are marked with arrows. The lettering indicates parents of the family: father (F), mother (M) and siblings as further referred to in the text and tables; B-brother, S-sister; P stands for the patients (probands) enrolled in this study.

**Table 1 jpm-11-00304-t001:** Variants identified in the patients and described in the publication.

Gene	Mutation	Consequence (Amino Acid)	rs SNP	P2	P3	P4	P5
*ADAMTS13*	c.1342C>G	p.Gln448Glu	rs2301612	C/G	C/G	G/G	G/G
*CD46*	c.-652G>A	-	rs2796267	G/G	G/G	G/G	G/G
*CD46*	c.-366G>A	-	rs2796268	G/G	G/G	G/G	G/G
*CD46*	c.899-78G>A	-	rs1962149	A/A	A/A	G/A	A/A
*CD46*	c.1037+638A>G	-	rs859705	A/A	A/A	A/A	A/A
*CD46*	c.783T>C	-	rs7144	C/C	C/C	T/C	T/T
*CD46*	c.673+58A>G	-	rs4844390	G/G	G/G	G/A	G/G
*CD46*	c.1127+43T>C	-	rs11118580	C/C	C/C	C/T	C/C
*CD46*	c.946+23G>A	-	rs2724374	G/G	G/G	T/G	G/G
*CFH*	c.2808G>T	p.Glu936Asp	rs1065489	C/T	C/T	C/T	C/T
*CFH*	c.-331C>T	-	rs3753394	C/T	C/T	C/T	C/T
*CFH*	c.2016A>G	p.Gln672Gln	rs3753396	A/G	A/G	A/G	A/G
*CFH*	c.1204C>T	p.His402Tyr	rs1061170	TT	TT	TT	TT
*CFH*	c.921A>C	p.Ala307Ala	rs1061147	CC	CC	CC	CC
*C3*	c.941C>T	p.Pro314Leu	rs1047286	C/T	C/T	C/T	C/T
*C3*	c.304C>G	p.Arg102Gly	rs2230199	C/G	C/G	C/G	C/G
*C3*	c.3489+69G>A	.	rs11569511	G/G	G/G	A/A	G/G
*C3*	c.4311C>T	p.Ala1437=	rs7951	C/C	C/C	C/T	C/C
*C3*	c.3230+36C>T	-	rs389404	C/C	C/C	C/T	C/C
*C3*	c.1692G>A	p.Val564=	rs2230204	G/G	G/G	G/A	G/G
*C3*	c.504+27A>G	-	rs2547438	A/A	A/A	A/G	A/A
*DGKE*	c.889-37T>A	-	rs1457404667	T/T	T/T	T/A	T/T

The CFH and CD46 risk haplotypes are shaded in gray.

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
