# Peer review of "Compound Haplotype Variants in CFH and CD46 Genes Determine Clinical Outcome of Atypical Hemolytic Uremic Syndrome (aHUS)—A Series of Cases from a Single Family"

_jpm, 2021, doi:10.3390/jpm11040304_

Round 1

Reviewer 1 Report

In this revised version, authors addressed my concerns and made the corrections accordingly. Paper can be published in its present form.

Author Response

Response to Reviewer 1 Comments

There was no suggestion by the Reviewer. I spell checked as noted in the paragraph “English language and style”.

Reviewer 2 Report

The manuscript has been improved and I have only ninor comments

  1. The authors should list at the end of the discussion the strengths and limitations of their study including the information what is novel in their study
  2. The manuscript is a case study that presents a series of cases from a single family and that information should be underlined by the authors in the manuscript and in its title. The family-based study is not an appropriate description of the study type. 
  3. The information about ravalizumab at the end of the discussion need to be removed since it may cause of conflict of interest. Neither ravalizumab was used in the study nor there is any experience with its use in patients with aHUS with variants in both CFH and CD46 genes

Author Response

Response to Reviewer 2 Comments

Point 1: The authors should list at the end of the discussion the strengths and limitations of their study including the information what is novel in their study

Response 1: Information about significance and limitations was added to the discussion, lines: 471-476.

Point 2: The manuscript is a case study that presents a series of cases from a single family and that information should be underlined by the authors in the manuscript and in its title. The family-based study is not an appropriate description of the study type. 

Response 2: In the title and the abstract I added emphasis that the paper is about the analysis of several cases. Please see lines: 4, 20, 71.

Point 3: The information about ravalizumab at the end of the discussion need to be removed since it may cause of conflict of interest. Neither ravalizumab was used in the study nor there is any experience with its use in patients with aHUS with variants in both CFH and CD46 genes.

Response 3: As suggested by the reviewer, the information about ravalizumab has been removed from the discussion.

This manuscript is a resubmission of an earlier submission. The following is a list of the peer review reports and author responses from that submission.

Round 1

Reviewer 1 Report

This is a case series reported within one family describing genetic variants leading to complement-mediated TMA (aHUS). Although the case series is interesting the findings are not novel and compound haplotype variants in CFH and CD46 genes have been well characterized and reported in the literature in large cohorts, e.g. doi: 10.1681/ASN.2018070759. Therefore this manuscript and family study does not add to our knowledge on the etiology of aHUS

Author Response

Response to Reviewer 1 Comments

Point 1: This is a case series reported within one family describing genetic variants leading to complement-mediated TMA (aHUS). Although the case series is interesting the findings are not novel and compound haplotype variants in CFH and CD46 genes have been well characterized and reported in the literature in large cohorts, e.g. doi: 10.1681/ASN.2018070759. Therefore this manuscript and family study does not add to our knowledge on the etiology of aHUS 

Response 1: Thank you for your comments. The publication that the reviewer cited was added as a citation in lines 586-588. In the manuscript presented here, we were interested in presenting the family history of the disease. Healthy family members were also analyzed and results are presented in the supplement. We believe that the publication is relevant as a familial analysis of aHUS.

Reviewer 2 Report

Dear authors,

The manuscript describes an impressive history of a family with aHUS and combined CD46/CFH mutation. The genotype-fenotype variation within one family is emphasized. This is not new. However, the family is well described, also in the context of limited availability of mutation analysis.

What I found confusing is that patient B1 is not called P1, because this is a patient too.

Furthermore I would prefer a table or figure with detailed genetic and clinical information of the patients.

Author Response

Response to Reviewer 2 Comments

Point 1: I found confusing is that patient B1 is not called P1, because this is a patient too.

 Response 1: Thank you for your comments. I corrected the nomenclature of the patients, please see Figure 1 and Table 1. That would be more readable.

Point 2: I would prefer a table or figure with detailed genetic and clinical information of the patients.

Response 2: I must admit that I found it impossible to address as crucial data have been already presented in Table 1 and Figure 1. For more details, please see the descriptions of the cases.

Reviewer 3 Report

In the current study, Furmańczyk-Zawiska et al. performed genetic analysis on five members of the same family affected by atypical haemolytic uraemic syndrome (aHUS) and observed a highly variable disease penetrance among those patients. While the study is quite satisfactory and results were presented in an organized fashion, there are several concerns need to be addressed to increase the value of the paper for the reader. Also, the English quality sometimes undermines the understanding of the text (please see the minor points).

Major concerns:

1) Authors mentioned about various bioinformatics tools in Methods section such as Trimmomatic, FastQC, Picard, ABRA, BCFtools, SnpSift, SnpEff, and VariantEffect. It would be good to see the vendors providing those or citations introducing these algorithms. The vendor information for  VIDITEST is also missing.

2) Please use the same nomenclature for ‘grams’ and ‘liters’ throughout the manuscript for the sake of consistency (e.g., HGB and PLT were presented with the units of g/dL and G/l, respectively).

3) In Materials and Methods, authors could include information about eculizumab, the recombinant humanized monoclonal antibody. Further in the manuscript (e.g., in the Results section), they could provide information about its working mechanism, optimal length of treatment and effectiveness.

4) In Figure 1, the siblings were listed in chronological order. Therefore, B4 needs to be placed in front of S3. Also, in the legend of the figure, please update the second sentence as the following:

- Symbol for deceased individual is stricken through with a forward-leaning line >> The deceased individual (B1) is depicted with a diagonal line through it.

5) Authors need to cite the related reference papers after the following sentences:

- In a French cohort of children … (pg. 8)

- On the contrary to our findings … (pg. 8)

- According to previously published data … (pg. 9)

6) In Discussion, authors could discuss

- about the age-relatedness of complement-mediated aHUS;

- whether the aHUS was causative of predisposing (i.e., requiring additional triggers such as other genetic factors, infections, etc.);

- about other effective therapies rather than eculizumab treatment.

Minor points:

1) Please correct the following typos:

- CHF functional region >> CFH functional region (pg. 2)

- manufacturaer’s >> manufacturer’s (pg. 3)

- Coomb’s >> Coombs’ (pg.s 3, 5 and 6)

- erythropoetin >> erythropoietin (pg. 5)

- Simpsone-Methode >> Simpson-Method (pg.s 5 and 6)

- cardiomiopathy >> cardiomyopathy (pg. 6)

- curently >> currently (pg. 6)

- disease14 >> disease [14] (pg. 6)

2) Please make the following changes in the related sentences as suggested:

- M220 Covaris ultrasonicator >> M220 focused-ultrasonicator (Covaris) (pg. 3)

- verified on the 2200 Tapestation >> verified on the 2200 Tapestation nucleic acids system (pg. 3)

- Illumina NextSeq500 >> NextSeq 500 sequencing system (Illumina) (pg. 3)

- bcl and fastq files >> bcl and fastq file formats (pg. 3)

- Trimmomatic 0.35 and FastQC >> Trimmomatic 0.35 and FastQC algorithms (pg. 3)

- (HCT) 18% >> (HCT) was 18% (pg. 3)

- (WBC) 13.2 G/l >> (WBC) was 13.2 G/l (pg. 3)

- C3 68,5 mg/dL >> C3 was 68,5 mg/dL (pg. 5)

- despite of >> despite (pg. 5)

- decreasing of EF was again (50%) observed >> decrease of EF (50%) was observed again (pg. 6)

- C3 68,6 mg/dL >> C3 was 68,6 mg/dL (pg. 6)

- renal function parameters and LDH decreased were observed >> decreased renal function parameters and LDH were observed (pg. 6)

- associated with slightly changes >> associated with slight changes (pg. 7)

- are shaded gray >> are shaded in gray (Table 1, pg. 7)

- Analyzing familial pedigree it has demonstrated that >> Analysis of familial pedigree demonstrated that (pg. 8)

- could be consider >> could be considered (pg. 8)

- estimated less than 20% >> estimated to be less than 20% (pg. 9)

- over >10 years >> over 10 years (pg. 9)

- SCr 1 mg/dL and 1.3 mg/dL >> 1.0 and 1.3 mg/dL SCr (pg. 9)

Author Response

Response to Reviewer 3 Comments

Point 1: Authors mentioned about various bioinformatics tools in Methods section such as Trimmomatic, FastQC, Picard, ABRA, BCFtools, SnpSift, SnpEff, and VariantEffect. It would be good to see the vendors providing those or citations introducing these algorithms. The vendor information for  VIDITEST is also missing.

Response 1: Thank you for your comments. I added a new reference papers, please see lines 550-571. The vendor for VIDITEST was VIDIA, which I added in line 84.

Point 2: Please use the same nomenclature for ‘grams’ and ‘liters’ throughout the manuscript for the sake of consistency (e.g., HGB and PLT were presented with the units of g/dL and G/l, respectively).

Response 2: I have corrected the nomenclature for grams and litres in the following lines: 132, 133, 134, 169, 263.

Point 3: In Materials and Methods, authors could include information about eculizumab, the recombinant humanized monoclonal antibody. Further in the manuscript (e.g., in the Results section), they could provide information about its working mechanism, optimal length of treatment and effectiveness.

Response 3: I have included more information about eculizumab in lines 76-78, 281-282 and 458-462.

Point 4: In Figure 1, the siblings were listed in chronological order. Therefore, B4 needs to be placed in front of S3. Also, in the legend of the figure, please update the second sentence as the following: - Symbol for deceased individual is stricken through with a forward-leaning line >> The deceased individual (B1) is depicted with a diagonal line through it.

Response 4: Figure 1 has been corrected, the siblings are listed in chronological order and the legend of the figure has been changed, please see lines 362-363.

Point 5: Authors need to cite the related reference papers after the following sentences:

- In a French cohort of children … (pg. 8)

- On the contrary to our findings … (pg. 8)

- According to previously published data … (pg. 9)

Response 5: The related reference papers have been included in lines 437, 442, 443, 448.

Point 6: In Discussion, authors could discuss

- about the age-relatedness of complement-mediated aHUS;

- whether the aHUS was causative of predisposing (i.e., requiring additional triggers such as other genetic factors, infections, etc.);

- about other effective therapies rather than eculizumab treatment.

Response 6: The Discussion has a new paragraph about :

  • age-relatedness of aHUS, please see lines 420-426
  • potential precipitating factors have been already discussed, please see lines 402-404 and 424-426
  • other effective therapies, please see lines 458-462.

Point 7: Please correct the following typos:

CHF functional region >> CFH functional region (pg. 2)

- manufacturaer’s >> manufacturer’s (pg. 3)

- Coomb’s >> Coombs’ (pg.s 3, 5 and 6)

- erythropoetin >> erythropoietin (pg. 5)

- Simpsone-Methode >> Simpson-Method (pg.s 5 and 6)

- cardiomiopathy >> cardiomyopathy (pg. 6)

- curently >> currently (pg. 6)

- disease14 >> disease [14] (pg. 6)

Response 7: The following types have been corrected, please see lines in the order of reviewer`s suggestions: 56, 84, 263, 283, 280, 288, 315.

Point 8: Please make the following changes in the related sentences as suggested:

- M220 Covaris ultrasonicator >> M220 focused-ultrasonicator (Covaris) (pg. 3)

- verified on the 2200 Tapestation >> verified on the 2200 Tapestation nucleic acids system (pg. 3)

- Illumina NextSeq500 >> NextSeq 500 sequencing system (Illumina) (pg. 3)

- bcl and fastq files >> bcl and fastq file formats (pg. 3)

- Trimmomatic 0.35 and FastQC >> Trimmomatic 0.35 and FastQC algorithms (pg. 3)

- (HCT) 18% >> (HCT) was 18% (pg. 3)

- (WBC) 13.2 G/l >> (WBC) was 13.2 G/l (pg. 3)

- C3 68,5 mg/dL >> C3 was 68,5 mg/dL (pg. 5)

- despite of >> despite (pg. 5)

- decreasing of EF was again (50%) observed >> decrease of EF (50%) was observed again (pg. 6)

- C3 68,6 mg/dL >> C3 was 68,6 mg/dL (pg. 6)

- renal function parameters and LDH decreased were observed >> decreased renal function parameters and LDH were observed (pg. 6)

- associated with slightly changes >> associated with slight changes (pg. 7)

- are shaded gray >> are shaded in gray (Table 1, pg. 7)

- Analyzing familial pedigree it has demonstrated that >> Analysis of familial pedigree demonstrated that (pg. 8)

- could be consider >> could be considered (pg. 8)

- estimated less than 20% >> estimated to be less than 20% (pg. 9)

- over >10 years >> over 10 years (pg. 9)

- SCr 1 mg/dL and 1.3 mg/dL >> 1.0 and 1.3 mg/dL SCr (pg. 9)

Response 8: The following types have been corrected, please see lines in the order of reviewer`s suggestions: 98-99, 100, 106-107, 110, 112-113, 133, 134, 257, 284, 296, 302, 352, 372-373, 417, 446, 451, 453, 454.